# Social perception of mesocarnivores within hunting areas differs from actual species abundance

**Bruno D. Suárez-Tangil[1]***, **Álvaro Luna[2]**

**1** Universidad Europea de Madrid, Madrid, Spain, **2** Department of Health Sciences, Facuty of Biomedical and Health Sciences, Universidad Europea de Madrid, Villaviciosa de Odón, Madrid, Spain

* bdstangil@hotmail.com

## Abstract

Analyzing how similar social perception and ecological field data are might help identify potential biases in identifying and managing human-carnivore conflicts. We analyzed the degree of similarity between the perceived and field-measured relative abundance to unveil whether attitude towards carnivores of two groups of stakeholders, namely hunters and other local people, is underpinned or it is instead biased by alternative factors. Our results indicate that, in general, mesocarnivore perceived abundances were generally different to actual species abundance. We also found that the perceived abundance and attributed damage to small game species were related with respondents' ability to identify the carnivore species. We underline the existence of bias and the need to increase people knowledge on species distribution and ecological characteristics before adopting decisions when managing human-carnivore conflicts, especially for stakeholders that are directly involved in.

**Data Availability Statement:** Dataset is now published in figshare repository with open access: Suárez de Tangil, Bruno (2023): PONE-D-22-29695: Responses to visual survey and

## Introduction

Human-wildlife conflicts are one of the most widespread threats facing wildlife species around the world [1]. This issue encompasses a huge diversity of situations and species, including the depredation upon livestock or game species, the damages on crops or stored food, the disturbances and attacks upon humans, the risk of disease transmission, or the allocation of large areas for biodiversity conservation instead of other extractive purposes [2–6]. Although there are multiple examples of human-carnivore coexistence (e. g., [7]), carnivores are frequently involved in human-wildlife conflicts. Human-carnivore conflicts arise when people consider that carnivore species damage livestock [8] prey on game species [9], or represent a threat to humans [10]. Moreover, the presence of carnivores across different ecosystems [11] entails the distribution of these human-carnivore conflicts to many different parts of the world.

The psychosocial framework of the conflict between carnivores and humans has unveiled the importance of the perceived risks and benefits associated with the presence of carnivores, which do not necessarily rely on logical cost-benefit analyses [12]. Indeed, the negative social perception of carnivores is not often in line with actual species behavior (e.g. [13]), and it is

questionnaire and field data. figshare. Dataset.
https://doi.org/10.6084/m9.figshare.22333120.v3.

**Funding:** Bruno D. Suárez-Tangil received the grant
Jorge G. Casanovas 2020 from the Terrestrial
Carnivores Group of the Spanish Society for the
Conservation and Study of Mammals (SECEM,
https://www.secem.es/).

**Competing interests:** The authors have declared
that no competing interests exist.

usually biased by social and cultural factors, economic considerations, personal experiences, and historical anecdotes [14]. Nevertheless, negative perception of carnivores can be minimized applying educational efforts emphasizing their positive impacts [12], such as benefits from wildlife tourism, commercial activities, regional and product marketing, game population control, or cultural heritage and identity [15]. In this context, to solve conflicts between humans and carnivores it might be necessary to apply management strategies that consider, apart from the underlying social context [16], the extent of the influence of those factors affecting individual preception. Specifically, management strategies should also consider whether objective evidences mainly underpin social perception of carnivores [17] or it is instead primarily shaped by alternative factors [18].

Human-carnivore conflicts entail divergent positions within society, which increases tension between different stakeholders and, therefore, hinder the adoption of appropriate decisions [19]. Recreational hunting represents an example of how different sectors of society can have different values, ethical acceptability, and perceptions about carnivores. Hence, analyzing the degree of similarity between objective evidences and individual perception, as well as differences in the perception of carnivores between hunters and non-hunters, might help identifying potential biases in the social perception of carnivores and, therefore, provide valuable information for future management strategies. In the Iberian Peninsula, some conflicts between humans and carnivores such as the Iberian wolf and the brown bear are widely known, mainly related with the depredation on livestock and beehives, respectively [20]. However, medium-sized carnivores, hereafter mesocarnivores, are also a source of conflict, especially for hunters [21], who highlight mortality of small game species by depredation as their main concern [22]. In Spain, recreational hunting is one of the most important activities carried out in rural environments [22, 23], representing a relevant proportion (0.3%) of the Gross Domestic Product and helping to maintain almost 200,000 employments (1% of people employed).

In order to provide useful information prior to the establishment of proper management strategies, field studies assessing mesocarnivore relative abundance across hunting areas (e. g., [21]) are fundamental. Moreover, interviews addressing questions related with predator control, the role of public administration, and the compatibility among hunting and nature conservation may uncover factors potentially influencing stakeholders' attitude towards carnivore [14], which is also critical. Nonetheless, few attempts have been made to integrate both approaches, comparing the results obtained from field studies assessing mesorcanivore relative abundance and the collection of individual perception of species distribution through surveys. Hence, our aim is fivefold. Firstly, we carried out surveys to investigate the opinion of two groups of stakeholders, namely hunters and other local people, about several aspects associated with the management of human-carnivore conflicts: predator control, the active involvement of public administration in promoting human-carnivore coexistence, and the compatibility among hunting and nature conservation. Secondly, we tested the ability of hunters and other local people to correctly identify the species and, therefore, analyze whether their responses were based in actual knowledge of species characteristics. Thirdly, we studied people perception about the relative abundance of mesocarnivores and small game species occurring within seven small-game hunting states, hereafter hunting areas. Fourthly, we quantified, through field samplings, the relative abundance of mesocarnivores and small game species occurring within the hunting areas. Finally, we analyzed the degree of similarity between the perceived and the measured relative abundance to unveil whether attitude towards carnivores is underpinned or it might be biased by alternative factors.

## Materials and methods

### Study area

We carried out the study across seven hunting areas distributed throughout the Autonomous Region of Extremadura, western Spain (Fig 1). The climate is temperate Mediterranean with rainfall occurring mostly during the hunting season [24], i. e. from October to January (240 mm; 53% of annual precipitation). Three different groups of hunting areas can be defined according to the vegetation cover and the main land uses. First, Mengabril (MEN) and Sierra de Fuentes (SFU) are agricultural lands occupied by cereal crops and a very scarce shrubland or forest cover (principally cork oaks and olive trees). Robledo (ROB) and Serrezuela 3 (SER)

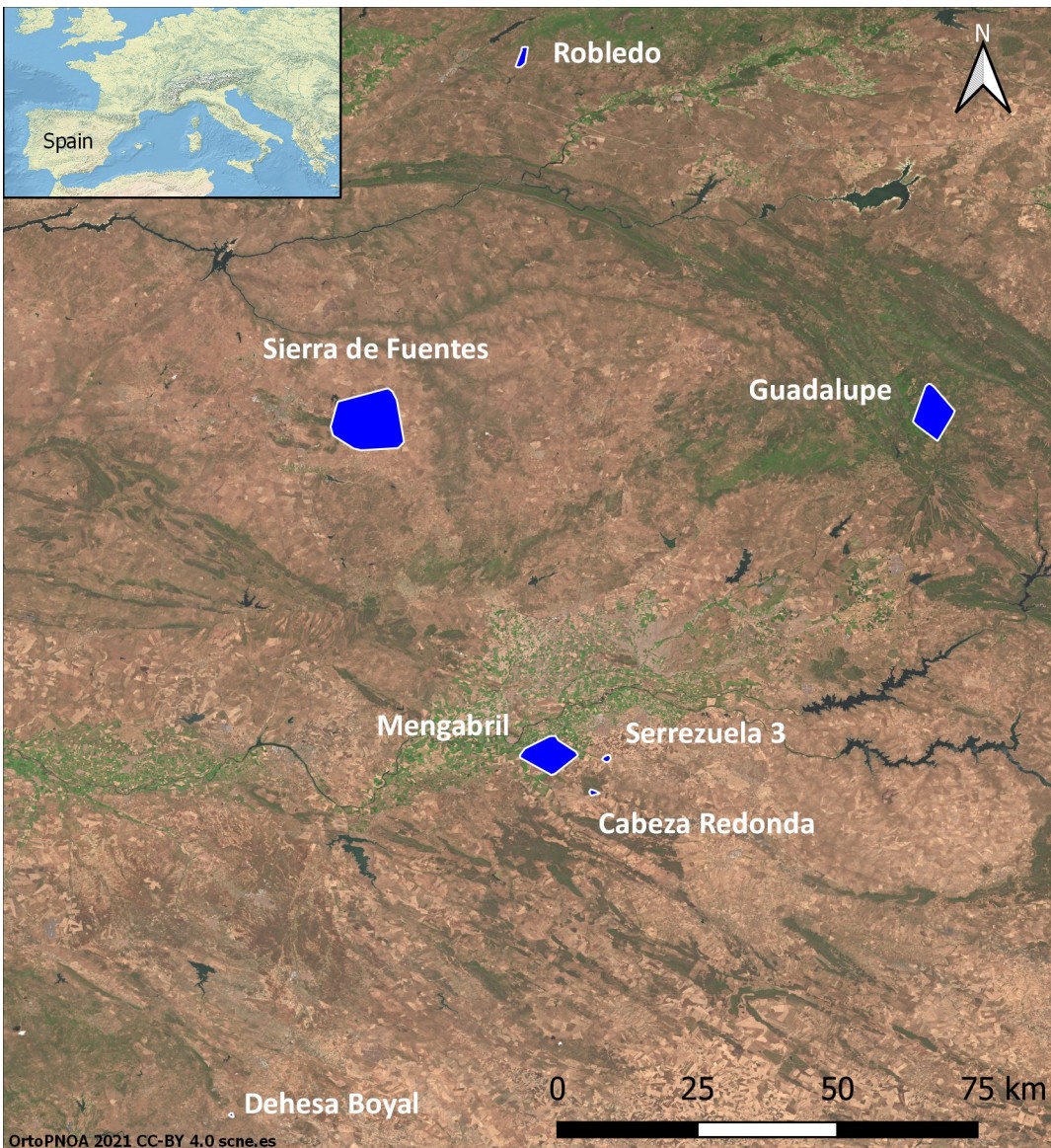

**Fig 1. Minimum convex polygons delimiting the hunting areas surveyed throughout the Autonomous Region of Extremadura, western Spain.** Delimitation between municipalities, provinces, and Autonomous Regions, represented with thin to thick lines, are shown for reference. Sources: OrtoPNOA 2021 CC-BY 4.0 scne.es and Natural Earth (public domain): http://www.naturalearthdata.com/.

are characterized by a patchy oak forest dedicated to recreational activities and/or livestock grazing. Finally, a mosaic of dense forest patches (primarily coniferous and oak forest), shrubland areas, and agricultural lands (mainly olive groves) comprises Cabeza Redonda (CRE), Dehesa Boyal (DBO), and Guadalupe (GUA).

## Data collection

**Social perception.** Considering the potential reluctance of people to collaborate in written questionnaires and the possible bias associated to this approach [25], we developed a visual approach used to study biodiversity perceptions in different countries and scenarios [26, 27]. We created a plate with color images of five mesocarnivore species distributed across the study area (S1A Fig): the red fox (*Vulpes vulpes*), the stone marten (*Martes foina*), the Eurasian badger (*Meles meles*), the common genet (*Genetta genetta*), and the Egyptian mongoose (*Herpestes ichneumon*). Likewise, we designed another plate with the most important small game species hunted in the area (S1B Fig): the European rabbit (*Oryctolagus cuniculus*), the Iberian hare (*Lepus granatensis*), and the red-legged partridge (*Alectoris rufa*). These species differed in natural size and anatomical features, but we depicted them with the same size on the plate to avoid biases with respect to visibility or conspicuousness. All the illustrations used for the surveys were created by the same author, and were initially designed to help identifying the species employing field guides [28, 29]. We employed images with the head of the animals to the left and, as far as possible, with similar position, to avoid specific traits of the image to have an effect on the choice of the respondents.

During each encounter, we asked each respondent to say the correct name of the species depicted in the first plate. We noted each name given by the respondents, even when they said an incorrect name or they did not give an answer. For mesocarnivore species, respondents were asked to evaluate, from 1 to 4, how abundant they consider each of the species in the corresponding hunting area (1 being very scarce or absence and 4 very abundant). Similarly, respondents also had to assess, from 1 to 4, the attributed damage caused to small game species across the hunting area (being 1 not conflictive for hunting purposes and 4 very conflictive). In the second plate, which represents the small game species, we only asked the respondents for the abundance of the species represented.

After the visual part of the survey, we followed with a traditional questionnaire that allowed us to obtain a more complete understanding of the social perception. Specifically, the questions proposed were:

1. "Do you think it is necessary to control the populations of the mesocarnivores represented in the plates?";

2. "Are there any other animal that you think can be harmful for small game species and, therefore, should be controlled beyond the ones in the picture?";

3. "Do you think hunting is compatible with nature conservation?"; and

4. "Would you consider as positive whether the administration develop activities and events, mainly addressed to young people, to promote the harmonisation of hunting and nature conservation?".

Regarding the question 2, we noted all the species given by the respondents. Finally, we consider the potential effect of gender and age in individual perception [27]. Thus, we noted the gender of the respondent, asked the respondents age (categorized in specific ranges), and questioned whether they lived nearby the hunting area in which we conducted the survey.

We conducted the surveys in two pre-defined groups of people: hunters and other local people, which describes people inhabiting villages nearby the hunting areas or that were

frequent users with other recreational purposes. Due to the limited number of people available in the area, the surveys were made to all suitable subjects older than 15 encountered across the towns located nearby the hunting areas. Surveys were conducted from October 2020 to January 2021. Despite we initially made the surveys in person, attending to the movement restrictions due to the COVID-19 crisis, part of the surveys were distributed among the potential respondents by collaborators. Once the surveys were completed, they sent back the surveys by e-mail. Given that our study was non-interventional, no ethical approval was sought. We only included the answers from those respondents who specifically provided verbal consent.

### Field sampling

In each hunting area, we placed a 4-km$^2$ square sampling unit on the Universal Transverse Mercator projection grid (Fig 2). Given the different surface occupied by the different hunting areas (ranging from 2.93 to 62.66 km$^2$, Table 1), we increased searching effort per hunting area according to its surface. Specifically, each 12-km$^2$ increase in surface area represented, approximately, an increase of one sampling unit (Fig 2). Hence, we placed one sampling unit across CRE (2.93 km$^2$), DBO (4.65 km$^2$), and SER (6.03 km$^2$), two sampling units across ROB (17.14

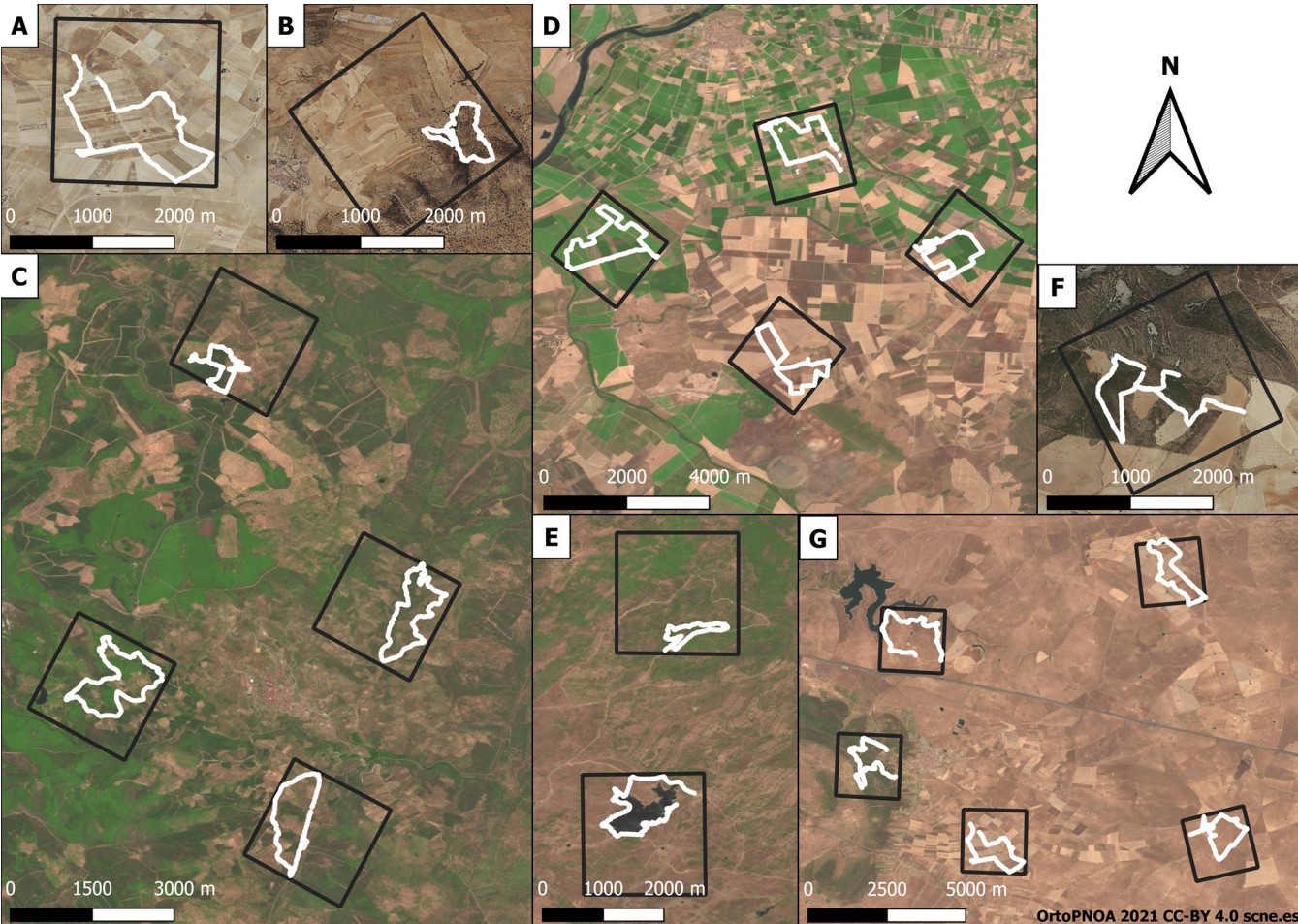

**Fig 2.** Distribution of 4-km$^2$ sampling units (squares) among the seven hunting areas (A-F). Within each sampling unit, the trajectory followed during the field sampling is represented. A) Cabeza Redonda (CRE); B) Dehesa Boyal (DBO); C) Guadalupe (GUA); D) Mengabril (MEN); E) Robledo (ROB); F) Serrezuela 3 (SER); G) Sierra de Fuentes (SFU). Source: OrtoPNOA 2021 CC-BY 4.0 scne.es.

**Table 1. Hunting area surface, distance covered, and effort applied during field sampling.**

| Hunting area | Surface (km²) | Survey length (km) | Sampling effort (min) |
|---|---|---|---|
| Cabeza Redonda | 2.93 | 5.86 | 120 |
| Dehesa Boyal | 4.65 | 3.79 | 120 |
| Guadalupe | 40.62 | 25.63 | 480 |
| Mengabril | 34.88 | 32.85 | 480 |
| Robledo | 17.14 | 11.90 | 240 |
| Serrezuela | 6.03 | 10.81 | 120 |
| Sierra de Fuentes | 62.66 | 34.94 | 600 |

km²), four sampling units across GUA (40.62 km²) and MEN (34.88 km²), and five sampling units across SFU (62.66 km²). In hunting areas with two or more sampling units, they were 2 km apart from each other to promote spatial independence between them. The regular performance and high efficiency of track and scat surveys have been proven in other Mediterranean regions with similar land covers [30, 31]. Therefore, we simultaneously carried out track surveys and scat surveys to estimate the relative abundance of mesocarnivores and small game species within each sampling unit. We also recorded opportunistic sightings. The mean ± SD length covered during field sampling was 7.0 ± 1.6 km (n = 18, Table 1). Track features were unique for species occurring in the region, and attributes of scats differ among target species. Hence, the distinctive characteristics of tracks and scats eased species identification across the study area. Fieldwork was performed only by experienced observers to minimize the probability of sign misidentification. However, any doubtful assignment of signs to species was discarded to avoid false positives. Transects within sampling units were conducted by two observers and the sampling effort was standardized at 120 minutes. Field sampling lasted 11 weeks, and both fieldwork and surveys were fully conducted during the hunting season (from October 2020 to January 2021).

## Statistical analyses

We employed generalized linear mixed models (GLMM) to analyse whether the ability to identify mesocarnivore species was different attending to respondent's gender, age, or their identification as hunters. As we noted the correct (1) or wrong (0) identification of mesocarnivore species, we considered a binomial distribution of the response variable for the analysis. To analyse whether social perception on a) mesocarnivore relative abundance, b) mesocarnivore damage on small game species, and c) small game species relative abundance, was different depending on respondent's gender, age, ability to identify the species, and their identification as hunters, we also employed GLMM. However, as both relative abundance and attributed damage were recorded as an index with four possible responses (ranging from 1 to 4), we considered a multinomial distribution of the response variables. Finally, we also used GLMM to analyse whether questionnaire responses depended on respondent's gender, age, or their identification as hunters. As we formulated questions to obtain "yes" (1) or "no" (0) answers, we employed the same procedure as with the species identification, and considered a binomial distribution of the response variable for the analysis. We used GLMM to control for environmental heterogeneity associated with the hunting areas, considering Hunting area as a random variable.

We used Fisher's exact tests to analyse whether the relative abundance of mesocarnivores and small game species recorded during field samplings were different among the hunting areas. Lastly, we analyze the degree of disparity between objective evidences (field data) and individual perception (survey data) and, therefore, the existence of bias. Specifically, we

estimated the Gower's distance to compare the relative abundance perceived by respondents and the relative abundance registered during track and scat surveys, and opportunistic sightings. To do this, we first computed the total number of responses obtained from each level of relative abundance (ranging from 1 to 4) and calculated the sum in proportion to the maximum potential value (i. e. the value in case all respondents from a hunting area would respond 4). Once the proportion was estimated, we assigned a perceived relative abundance to each hunting area considering that i) when the sum correspond to a proportion lower than 25.00% of maximum potential value, the value was 1; ii) when the sum correspond to a proportion between 25.01% and 50.00% of maximum potential value, the value was 2; iii) when the sum correspond to a proportion between 50.01% and 75.00% of maximum potential value, the value was 3; and iv) when the sum correspond to a proportion between 75.00% and 100.00% of maximum potential value, the value was 4. For example, the sum of the relative abundance values provided by the respondents for the common genet in ROB was 17, which represents the 60.71% of the maximum potential value (seven respondents x 4 = 28). Therefore, the relative value perceived by respondents that was assigned to common genet in ROB was 3. Likewise, in the case of field samplings, we established an index of relative abundance by calculating the proportion of records found in each hunting area with respect to the maximum observed value (i. e. the maximum number of records found in a hunting area). For example, the records of stone marten across SFU was 8, which represents the 32.00% of the maximum value observed (GUA, n = 25). Thus, the relative abundance value that was assigned to stone marten in SFU was 2. Once the index of relative abundance was computed for both social perception and field sampling, we estimated the Gower's distance, which ranged from 0 to 1. Values close to 0 represent minor dissimilarities between the relative abundance perceived by the respondents and the relative abundance recorded during field samplings, whereas values close to 1 represent substantial dissimilarities.

All statistical analyses were carried out using R software [32]. GLMM with binomial data distribution were carried out employing the glmer function from the "lme4" package [33], whereas GLMM with multinomial data distribution were carried out using the mblogit function from "mclogit" package [34]. Finally, Gower's distance was estimated employing gower. dist function from the "StatMatch" package [35].

## Results

### Socio-demographic factors and attitude towards mesocarnivores

We completed 119 surveys across the seven hunting areas. Two respondents (1.68%) were from CRE, 16 (13.45%) from DBO, 25 (21.01%) from GUA, 17 (14.29%) from MEN, seven (5.88%) from ROB, 16 (13.45%) from SER, and 36 (30.25%) from SFU (S2A Fig). Most respondents (95.80%) were from localities near the hunting areas, 74 (62.18%) were older than 41 years old (S2B Fig), 106 (89.08%) were men (S2C Fig), and 74 (62.18%) were identified as hunters (S2D Fig). 116 respondents (97.48%) agreed with the active involvement of the public administration, to explain, especially to young people, the importance and compatibility of hunting and nature conservation. Indeed, 111 respondents (93.28%) said that hunting was completely compatible with nature conservation, being hunters more convinced than other local people (Coefficient = 2.47, SE = 1.10, P = 0.025). Nevertheless, 102 respondents (85.71%) agreed with mesocarnivore control and 79 (66.39%) explicitly mentioned additional species they perceived they should be controlled (S2E Fig). Among the noted species, carnivores such as the feral cat (*Felis catus*) was mentioned by 43 respondents (54.43%), the domestic dog (*Canis lupus familiaris*) by seven (8.86%), and the European polecat (*Mustela putorius*) by another respondent (1.27%). The wild boar was also noted by 22 respondents (27.85%).

Furthermore, avian species such as white storks (*Ciconia ciconia*), magpies (*Pica pica*), and "birds of prey" were mentioned by 26 respondents (32.91%). Given the low proportion of surveys answered by women (10.92%), we did not consider the effect of gender in the responses. Similarly, age group of 15–18 years old was excluded from analyses considering respondents' age due to the low sample size (n = 1).

### Identification of mesocarnivore species

Only 48 respondents (40.33%) adequately identified all mesocarnivore species, 30 respondents (25.21%) wrongly identified between 1 to 4 species, whereas three (2.52%) were not able to correctly identify any species (the remaining surveys, 31.94%, represent those in which the respondents did not provide an answer about the identification of some species, S1 Table). The red fox was identified accurately the most (97.48%), followed by the Eurasian badger (84.03%), the Egyptian mongoose (77.31%), the common genet (75.63%), and the stone marten (44.54%). Specifically, stone marten identification was mistaken by 23 respondents (76.67%), common genet and Egyptian mongoose by four respondents each (13.33%), Eurasian badger by three (10.00%), and red fox by one (3.33%). Among the mistaken responses, the stone marten was primarily confused with the Eurasian otter (*Lutra lutra*, 86.96% of the erroneous name given), the common genet with the Iberian lynx (*Lynx pardinus*, 50.00%), the Egyptian mongoose with the least weasel (*Mustela nivalis*, 75%), and the red fox with the Iberian wolf (*Canis lupus signatus*, 100.00%). The Eurasian badger was confused with three taxa non-indigenous for Europe: the skunk (Mephitidae), the raccoon (Procyonidae), and the anteater (Myrmecophagidae). Except the red fox, hunters identified all mesocarnivores more successfully than other local people surveyed (Table 2). Conversely, the age of the respondents did not substantially affect the success of species identification (S2 Table).

### Social perception of species abundance and attributed damage

The mean (±SD) abundance of mesocarnivores perceived by respondents (index ranging from 1 to 4) was 3.64 (±0.71) for the red fox, 3.22 (±1.10) for the Egyptian mongoose, 1.96 (±1.04) for the Eurasian badger, 1.77 (±0.97) for the common genet, and 1.59 (±0.86) for the stone marten (Fig 3). We found that the perceived abundance was significantly different between hunters and other local people (S3 Table). Specifically, hunters perceived a higher abundance of Egyptian mongoose than other local people did, and a medium abundance of the Eurasian badger and common genet in comparison to other local people. Nevertheless, no significant differences were found between both groups in the case of the red fox, which was perceived as abundant, and the stone marten, which was considered rare. The mean (±SD) damage on small game species attributed to carnivores was 3.49 (±1.01) for the Egyptian mongoose, 3.36

**Table 2. Parameter estimates showing differences in ability to identify mesocarnivores between hunters and other local people.**

| Species | Coefficient | SE | P |
|---|---|---|---|
| *Stone marten* | **2.10** | **0.65** | **<0.01** |
| *Eurasian badger* | **3.90** | **1.12** | **<0.01** |
| *Common genet* | **3.56** | **0.69** | **<0.01** |
| *Egyptian mongoose* | **3.74** | **0.87** | **<0.01** |

The group 'other local people' is included in the intercept. Significant differences between hunters and other local people are marked in bold. Given that the red fox was correctly identified by most of respondents (97.48%), binomial GLMM did not converge and model output was omitted.

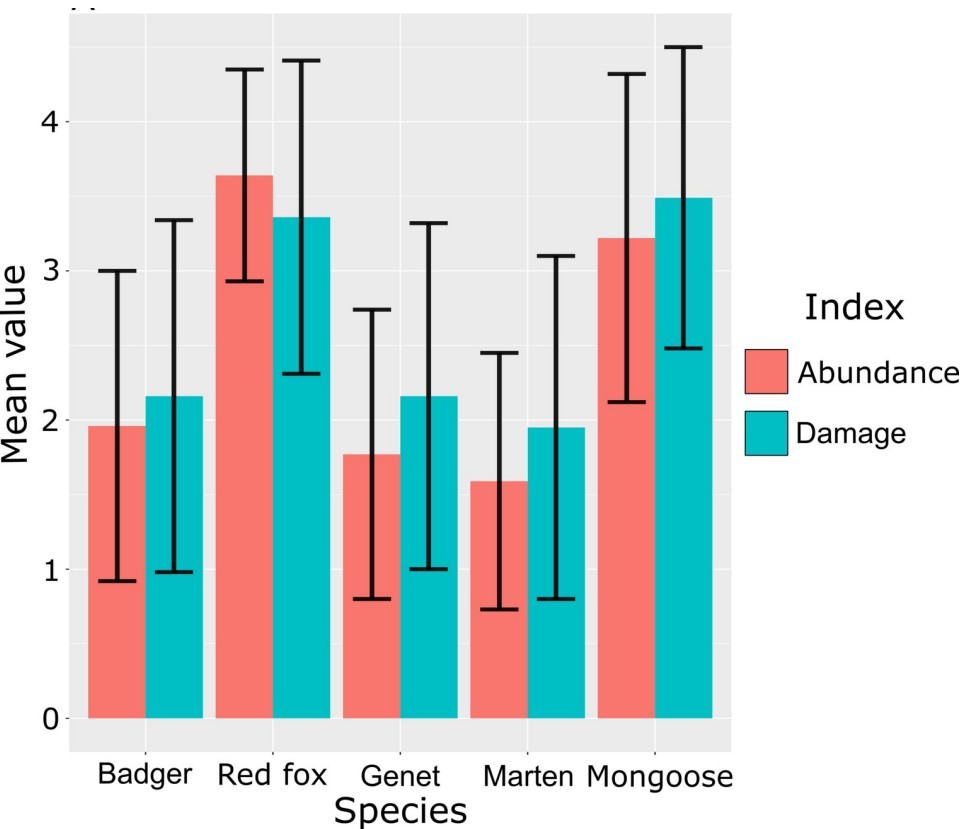

**Fig 3. Social perception.** Mean relative abundance of mesocarnivores and damage caused to small game species perceived by the respondents.

(±1.05) for the red fox, 2.16 (±1.18) for the Eurasian badger, 2.16 (±1.16) for the common genet, and 1.95 (±1.15) for the stone marten. In this case, we also found that the perceived damage caused to small game species by mesocarnivores was significantly different between hunters and other local people (S3 Table). Hunters perceived a larger damage in the case of the red fox, the stone marten, the common genet, and the Egyptian mongoose. Conversely, hunters perceived that the damage caused to small game species by the Eurasian badger was medium-high. Additionally, we found that, in the case of the stone marten and the Egyptian mongoose, the perceived relative abundance was different among those respondents that knew the species from those who did not. Specifically, whereas stone marten was perceived as less abundant by those who correctly identified the species, the Egyptian mongoose was considered as more abundant. Also, the attributed damage to small game species was perceived as higher for the Egyptian mongoose by the respondents that adequately identified the species. Finally, regarding the small game species, the mean species abundance perceived by respondents was 2.01 (±1.08) for the European rabbit, 1.60 (±0.81) for the Iberian hare, and 1.97 (±0.86) for the red-legged partridge (S3 Fig). We found significant differences in the relative abundance of the red-legged partridge perceived by hunters in comparison to other local people (S4 Table), which described a medium-low abundance of the species in the hunting areas. However, we did not find significant differences between hunters and other local people for the European rabbit and the Iberian hare. Overall, no significant differences in the perception of the relative abundance of the species or the damage caused by the mesocarnivores were found between age groups (S5 and S6 Tables).

**Table 3. Relative abundance of mesocarnivore and small game species recorded during field samplings across each 4-km² sampling unit.**

| Hunting area | Mesocarnivores | | | | | Small game species | | |
|---|---|---|---|---|---|---|---|---|
| | Red fox | Stone marten | Eurasian badger | Common genet | Egyptian mongoose | European rabbit | Iberian hare | Red-legged partridge |
| *Cabeza Redonda* | **8** | **3** | **1** | **0** | **0** | **69** | **1** | **23** |
| *Dehesa Boyal* | **12** | **6** | **3** | **5** | **0** | **0** | **0** | **4** |
| *Guadalupe I* | 1 | 7 | 0 | 0 | 0 | 0 | 0 | 5 |
| *Guadalupe II* | 5 | 6 | 1 | 1 | 0 | 0 | 0 | 0 |
| *Guadalupe III* | 9 | 6 | 0 | 0 | 0 | 0 | 0 | 37 |
| *Guadalupe IV* | 18 | 6 | 4 | 0 | 0 | 0 | 0 | 0 |
| *Guadalupe* | **33** | **25** | **5** | **1** | **0** | **0** | **0** | **42** |
| *Mengabril I* | 3 | 0 | 0 | 0 | 0 | 16 | 0 | 3 |
| *Mengabril II* | 2 | 0 | 0 | 0 | 0 | 2 | 0 | 129 |
| *Mengabril III* | 2 | 0 | 0 | 0 | 0 | 0 | 0 | 59 |
| *Mengabril IV* | 2 | 0 | 0 | 0 | 1 | 15 | 0 | 79 |
| *Mengabril* | **9** | **0** | **0** | **0** | **1** | **33** | **0** | **270** |
| *Robledo I* | 4 | 1 | 0 | 1 | 1 | 1 | 1 | 2 |
| *Robledo II* | 10 | 1 | 1 | 1 | 2 | 0 | 7 | 12 |
| *Robledo* | **14** | **2** | **1** | **2** | **3** | **1** | **8** | **14** |
| *Serrezuela 3* | **4** | **0** | **1** | **0** | **1** | **675** | **3** | **27** |
| *Sierra de Fuentes I* | 4 | 0 | 0 | 0 | 0 | 1 | 2 | 2 |
| *Sierra de Fuentes II* | 5 | 0 | 0 | 0 | 0 | 1 | 2 | 1 |
| *Sierra de Fuentes III* | 6 | 0 | 0 | 0 | 0 | 25 | 41 | 16 |
| *Sierra de Fuentes IV* | 9 | 0 | 0 | 0 | 0 | 7 | 22 | 20 |
| *Sierra de Fuentes V* | 4 | 8 | 0 | 1 | 0 | 0 | 0 | 5 |
| *Sierra de Fuentes* | **28** | **8** | **0** | **1** | **0** | **34** | **67** | **44** |

Total relative abundance recorded across each hunting area is marked in bold.

## Field sampling

We found differences in the distribution of both mesocarnivore and small game species among the hunting areas (Table 3, S4 Fig). Indeed, the red fox and the red-legged partridge were the only species found across the whole study area. Regarding the other mesocarnivore species, we registered the occurrence of the stone marten, the Eurasian badger, and the common genet across five hunting areas (71.43%), whereas the Egyptian mongoose was registered only across three (42.86%). In the case of the remaining small game species, we registered the occurrence of the European rabbit across five hunting areas (71.43%), and the Iberian hare across four (57.14%). Considering the number of records registered for each mesocarnivore species, we found the highest relative abundance for the red fox (108), followed by the stone marten (44), the Eurasian badger (11), the common genet (9), and the Egyptian mongoose (5). Mesocarnivore relative abundance was significantly different across the hunting areas (Fisher's exact test p = 0.001). In the case of small game species, we found the highest relative abundance for the European rabbit (812), followed by the red-legged partridge (424), and the Iberian hare (79). The relative abundance of small game species was also different across the hunting areas (Fisher's exact test p < 0.001).

## Social perception vs. field samplings

We found dissimilarities between the relative abundance perceived by the respondents and the relative abundance registered during field surveys (Table 4, S7 Table). Overall, mesocarnivores

**Table 4. Dissimilarity between the relative abundance perceived by respondents and the relative abundance recorded during field sampling.**

| | Species | Social perception *vs*. field sampling | | |
| --- | --- | --- | --- | --- |
| | | Total | Hunters | Other local people |
| **Mesocarnivores** | *Red fox* | 0.57 | 0.50 | 0.72 |
| | *Stone marten* | 0.29 | 0.28 | 0.22 |
| | *Eurasian badger* | 0.29 | 0.44 | 0.17 |
| | *Common genet* | 0.33 | 0.39 | 0.22 |
| | *Egyptian mongoose* | 0.57 | 0.72 | 0.59 |
| **Small game species** | *European rabbit* | 0.60 | 0.58 | 0.39 |
| | *Iberian hare* | 0.48 | 0.56 | 0.19 |
| | *Red-legged partridge* | 0.76 | 0.67 | 0.75 |

We estimated dissimilarity in the relative abundance considering altogether the total of surveys, and separately the surveys answered by hunters, and the surveys answered by other local people.

Values close to 0 represent minor dissimilarities between the relative abundance perceived by the respondents and the relative abundance recorded during field samplings, whereas values close to 1 represent substantial dissimilarities between the relative abundance perceived by the respondents and the relative abundance recorded during field sampling.

and small game species were perceived more abundant than they were in the field. However, this perception was not uniform across species. In the case of mesocarnivore species, the red fox and the Egyptian mongoose were perceived almost twice as abundant as the stone marten, the Eurasian badger, and the common genet (Table 4, S7 Table). Conversely, despite small game species were also perceived as more abundant than they were in the field, differences were especially remarkable in the case of the European rabbit and the red-legged partridge (Table 4, S7 Table). We found that, in general, the dissimilarities between the perceived mesocarnivore abundance and the abundance observed in the field were lower for other local people than for hunters (Table 4, S7 Table). Specifically, in comparison to other local people, dissimilarities obtained for hunters were higher for all mesocarnivores except for the red fox. Similarly, dissimilarities obtained for hunters were higher for all small games species except for the red-legged partridge (Table 4, S7 Table).

## Discussion

Our results indicate that the relative abundance perceived by the respondents was not supported by the relative abundance measured during field samplings: hunters overestimated mesocarnivores relative abundance, whereas other local people overstimated small game species relative abundance. In the case of mesocarnivores, our results indicate that the perceived relative abundance may be biased to species whose circadian activity is mostly diurnal such as the Egyptian mongoose [36] or opportunistic species such as the red fox [37]. For small game species, and comparing the two groups of surveyed people, there is a greater similarity between the relative abundance perceived by hunters and the abundance measured. This can be justified by hunters' knowledge and interest about the presence of the small game species in a given area. Therefore, considering simultaneously the objective evidences collected through field samplings and the individual perception compiled through surveys, we underline the existence of biases. Some of the main factors influencing human-carnivore coexistence are education and knowledge [38]. Hence, we highlight the need of increasing knowledge for the society in general, and especially for stakeholders that are directly involved in human-carnivore conflicts such as hunters.

The greatest ability to identify the species showed in the plates was demonstrated by hunters. Nevertheless, our results also indicate that most of the respondents were not able to visually identify all the species, and that the perceived abundance and attributed damage were related with respondents' ability in the case of the Egyptian mongoose and the stone marten. Damage caused to small game species by mesocarnivores has been seldom quantified [39]. However, our results indicate that the average damage attributed to carnivore by respondents was considered low only for a small group on respondents. Additionally, in the case of the Egyptian mongoose, damage attributed was related with respondent's ability to identify the species. It is plausible that the proliferation of unverified information published on social networks and biased news in the media, which can lead to the spread of erroneous information and unjustified low tolerance [40], may have contributed to these results. Moreover, emotional, cultural, or personal considerations may primarily determine social perception of carnivores and their attributed damage to small game species [9], evidencing the need of information and education to deal with and avoid conflict [41]. Our results seem to reinforce the idea that the perceived relative abundance and the damage attributed to carnivores might be biased, and that they are, in most cases, independent from people's knowledge.

Our study also provides contrasting results between the positive attitude towards nature conservation, as a whole, and the negative attitude towards the conservation of mesocarnivores. In this sense, the respondents considered that hunting was compatible with nature conservation. However, people considered species lethal control as necessary. Regard this, control measures are frequently employed to mitigate human-wildlife conflicts [42], being especially applicable to control domestic or invasive alien species [43, 44]. Nevertheless, the use of these measures in hunting states is controversial. Restrictions on hunting species that are supposed to generate conflict are usually violated [45]. Predator control, nevertheless, promotes the imbalance of natural ecosystems and, paradoxically, produces effects contrary to those targeted when predator control is implemented [46, 47]. Moreover, long-term efficiency of predator control have been showed to be weak [9, 48]. Our results therefore suggest that more educational and outreach efforts are needed to explain the role of mesocarnivores across ecosystems and re-think on the long established idea that predator control is the only option to choice to solve human-carnivore conflicts.

The public interest in nature conservation and the engagement of public administrations in managing natural resources has significantly increased in the last decades [49, 50]. Our results may provide valuable information to public administration for the design of future management strategies. Specifically, we describe three interesting aspects that may be overlooked if social perception and objective evidences are considered independently when dealing with human-carnivore conflicts. Firstly, we demonstrated disparities between the actual species abundance and the perceived abundance. Hence, before designing management strategies, it should be essential to contrast to what extent the perception of stakeholders match with the actual abundance of the species source of conflict. Secondly, we found that, despite the ability to identify carnivore species was higher in hunters than in other local people, their perception might still be biased, and that it is, in most cases, independent from knowledge. Therefore, new efforts may be applied not only to increase common people knowledge about the species inhabiting natural environment, but also to explain the role of mesocarnivores across ecosystems, and their potential benefits instead of negative impacts. Finally, regardless the confidence on predator control as a management tool for the protection of small game species, respondents considered the active involvement of public administration to harmonize hunting and nature conservation as positive. This willingness might favour the development of the initial stages of strategies targeted to promote human-carnivore coexistence.

## Conclusions

Our results suggest that it is critical to consider social perception to unveil gaps in the management of human-carnivore conflicts. However, to avoid unjustified bias and adopt informed and appropriate resolutions, social perception should be considered in conjunction with a comprehensive knowledge of species distribution, which can only be achieved throughout field samplings. The perceived relative abundance of mesocarnivores and small game species were overestimated in comparison to the actual relative abundance registered during field samplings. Specifically, hunters overestimated the relative abundance of mesocarnivores, while other local people overestimated the relative abundance of small game species. Additionally, we found contrasting results between a positive attitude towards nature conservation, as a whole, and a negative attitude towards the conservation of mesocarnivore species, for which predator control is usually claimed. The negative attitude towards mesocarnivores, measured as attributed damage to small game species, is more evident for well-known species than for little-known species, and was especially remarkable in the case of hunters. Our results may provide valuable information for the design of future management strategies, as they describe three interesting aspects that may be overlooked if social perception and objective evidences are considered independently when dealing with human-carnivore conflicts: disparities between perceived and actual species abundance, biases in social perception derived from knowledge gaps in species identification, and positive willingness to harmonize hunting and nature conservation. Given that we focus on a single-case study, further research is required, including comparisons across case studies, the analysis of potential disparities in people perception among sites, or the study of potential changes in the degree of similarity among hunting areas. Additionally, without the restrictions derived from the COVID-19 pandemic, a more consistent survey replication and a larger sample size would be desirable. Nevertheless, our study clearly indicates that social perception should be considered in conjunction with a comprehensive knowledge of species distribution in order to avoid unjustified bias and adopt informed and appropriate decisions in the management of human-carnivore conflicts.

## Supporting information

**S1 Fig.** Plates with color images employed during the surveys representing A) the mesocarnivore species, and B) the small game species, considered in the study. 1) Red fox (*Vulpes vulpes*); 2) Stone marten (*Martes foina*); 3) Eurasian badger (*Meles meles*); common genet (*Genetta genetta*); 5) Egyptian mongoose (*Herpestes ichneumon*).
(PDF)

**S2 Fig. Socio-demographic factors.** A) Proportion of surveys carried out in each hunting area. B) Number of surveys completed by each age group. C) Proportion of surveys completed regarding the gender of the respondent. D) Proportion of surveys completed by hunters and other local people. E) Number of surveys in which respondents considered additional species that need to be controlled.
(PDF)

**S3 Fig. Social perception.** Mean relative abundance of small game species perceived by the respondents.
(PDF)

**S4 Fig.** Relative abundance of A) mesocarnivores, and B) small game species, recorded during field samplings. Grey bars represent the proportion of records attributed to the species registered (in columns) and the proportion of records found across each hunting area (in rows).

Light blue dots represent the relative abundance of each species across each hunting area. Differences in dot size represent between-species differences in relative abundance. CRE: Cabeza Redonda; DBO: Dehesa Boyal; GUA: Guadalupe: MEN: Mengabril; ROB: Robledo; SER: Serrezuela 3; SFU: Sierra de Fuentes.
(PDF)

**S1 Table. Parameter estimates showing differences in capacity of mesocarnivore identification between age groups.**
(PDF)

**S2 Table.** Parameter estimates showing differences in the perception of A) mesocarnivore relative abundance, and B) damage caused to small game species, between hunters and other local people.
(PDF)

**S3 Table.** Parameter estimates showing differences in the perception of A) mesocarnivore relative abundance, and B) damage caused to small game species, depending on the ability to adequately identify the species.
(PDF)

**S4 Table. Parameter estimates showing differences in the perception of the relative abundance of small game species between hunters and other local people.**
(PDF)

**S5 Table.** Parameter estimates showing differences in the perception of A) mesocarnivore relative abundance, and B) damage caused to small game species, between age groups.
(PDF)

**S6 Table. Parameter estimates showing differences in the perception of the relative abundance of small game species between age groups.**
(PDF)

**S7 Table.** Degree of similarity between the relative abundance perceived by A) the total of respondents, B) hunters, and C) Other local people and the relative abundance registered during the field samplings.
(PDF)

**S8 Table. Relative abundance of mesocarnivore and small game species recorded during field samplings across each sampling unit by each sampling method.**
(PDF)

## Acknowledgments

We thank D Prado, E Garzón, and J Mateos from the Sociedad Extremeña de Zoología (SEZ, https://sezextremadura.org/), S Pinto from Brutal (https://www.brutal.org.es/), and C Terán for assistance with fieldwork. We also thank many collaborators who granted permission to access the hunting estates, and three reviewers for providing helpful comments.

## Author Contributions

**Conceptualization:** Bruno D. Suárez-Tangil.

**Data curation:** Bruno D. Suárez-Tangil.

**Formal analysis:** Bruno D. Suárez-Tangil.

**Funding acquisition:** Bruno D. Suárez-Tangil.

**Investigation:** Bruno D. Suárez-Tangil, Álvaro Luna.

**Methodology:** Bruno D. Suárez-Tangil, Álvaro Luna.

**Project administration:** Bruno D. Suárez-Tangil, Álvaro Luna.

**Resources:** Bruno D. Suárez-Tangil.

**Software:** Bruno D. Suárez-Tangil.

**Supervision:** Bruno D. Suárez-Tangil.

**Validation:** Bruno D. Suárez-Tangil, Álvaro Luna.

**Visualization:** Bruno D. Suárez-Tangil, Álvaro Luna.

**Writing – original draft:** Bruno D. Suárez-Tangil.

**Writing – review & editing:** Bruno D. Suárez-Tangil, Álvaro Luna.

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
