## [Decision Letter · Decision Letter 0]

3 Jan 2023

PONE-D-22-29695Social perception of mesocarnivores within hunting areas differs from actual species abundancePLOS ONE

Dear Dr. Suárez de Tangil,

Thank you for submitting your manuscript to PLOS ONE. After careful consideration, we feel that it has merit but does not fully meet PLOS ONE’s publication criteria as it currently stands. Therefore, we invite you to submit a revised version of the manuscript that addresses the points raised during the review process.

We look forward to receiving your revised manuscript.

Kind regards,

Bogdan Cristescu

Academic Editor

PLOS ONE

Journal Requirements:

 "Bruno D. Suárez-Tangil received the grant Jorge G. Casanovas 2020 from the Terrestrial Carnivores Group of the Spanish Society for the Conservation and Study of Mammals (SECEM, https://www.secem.es/).

Bruno D. Suárez-Tangil and Álvaro Luna conceived and designed the study; Bruno D. Suárez-Tangil and Álvaro Luna collected the data during field samplings and carried out part of the surveys; Bruno D. Suárez-Tangil analysed the data; Bruno D. Suárez-Tangil led the drafting of the manuscript. " 

5. Please include your tables as part of your main manuscript and remove the individual files. Please note that supplementary tables (should remain/ be uploaded) as separate "supporting information" files

Additional Editor Comments:

Two reviewers have perused the manuscript and differ widely in their recommended decision (Reject vs. Accept). Reviewer 1 in particular has excellent comments and suggestions. I therefore invite you to address all the comments of the reviewers as well as my comments below in a revised manuscript.

Some of the targeted taxa are not mesocarnivore - they are small carnivores (e.g., de Satge et al. 2017 Oecologia 184(4):873-884). They should be classified accordingly throughout the manuscript.

European rabbit vs. Iberian hare - can you tell the difference from tracks?

I disagree that no studies have linked social and ecological data on carnivore distribution in conflict areas. There are many examples of socio-ecological systems and using traditional knowledge in conjunction with species distribution modelling for conflict-prone carnivores. Please add some citations to this effect.

It would be very useful to add information on size of hunting areas and survey effort in a table.

The calculation of the relative abundance does not seem to be the most appropriate - see for example comments from Reviewer 1. Please also consider changing the standardization of indices (see Reviewer 1's recommendations). Even better, if the data allow an occupancy analysis approach (incorporating detection probability), please replace the relative abundance indices calculations with occupancy analysis. A less favored alternative would be to keep the relative abundance indices, but to acknowledge their limitations.

Records should not be pooled across field methods (e.g., in Table 2).

Add scale bar (km) to panels in Fig. S2. This figure is important as it shows the sampling units and should be moved to the main manuscript. It shows a gradient of habitat types - from relatively intact habitats to agricultural areas - have you considered incorporating habitat types in the analysis as this will influence carnivore distribution?

Fig. S4 (panel A) is also important to include in the main manuscript.

Please credit the background satelite image in Fig. 1.

Reviewers' comments:

Reviewer's Responses to Questions

**Comments to the Author**

1. Is the manuscript technically sound, and do the data support the conclusions?

Reviewer #1: No

Reviewer #2: Yes

2. Has the statistical analysis been performed appropriately and rigorously? 

Reviewer #1: No

Reviewer #2: Yes

3. Have the authors made all data underlying the findings in their manuscript fully available?

Reviewer #1: No

Reviewer #2: Yes

4. Is the manuscript presented in an intelligible fashion and written in standard English?

Reviewer #1: No

Reviewer #2: Yes

5. Review Comments to the Author

Reviewer #1: I have reviewed the manuscript entitled: “Social perception of mesocarnivores within hunting areas differs from actual species abundance”. The manuscript covers a very important topic in wildlife management and conservation. I have several concerns regarding the underlying assumptions and methodology though. These concerns are raised below.

For the manuscript in question to make the claims it does, we need to know how reliable the so-called “real abundance” estimates are. This is because the study does not aim, for example, to compare two methods of detecting a species’ presence, but instead aims to examine how closely people’s perceptions of a species’ abundance match reality. Understandably, robust estimates of population abundance or density are not feasible for these species. Therefore, the use of a proxy is required. For the study in question, the authors use the number of detections (track, scat, or visual/opportunistic) as the basis for their relative abundance index. However, relative abundance indices that do not account for imperfect detection are known to be flawed measures that may produce biased or even spurious results. Since the underlying thesis of this study depends on the reliability of the field-sampling based abundance estimates, one must appreciate this limitation severely hinders the manuscript in question. On this criticism alone I would say the manuscript is dead-on-arrival. If I may ask, are the field data wholly unsuitable for occupancy analysis? Using occupancy probabilities derived from a model that accounts for imperfect detection to rank-order species would be better.

Moving beyond the quality of the data, there is the issue of how the authors produced the relative abundance scores. The approach used seems arbitrary and I can imagine the study’s results will probably be sensitive to the scoring system used. Currently, the authors sum up the number of detections within a given hunting area and then divide each area’s total by the total number of detections within hunting area with the most detections. I imagine the authors could just as easily divide each hunting area’s total by the total number of detections from all hunting areas combined and that would presumably yield different results (though perhaps not that different since most detections come from one or two hunting areas). Likewise, the authors could easily average the number of detections across surveys within each hunting area and use hunting area averages instead of hunting area totals. I’m sure we can think up countless ways to score a species’ relative abundance, but I bring up the use of averages specifically for a reason. By totaling the number of detections across multiple surveys, the authors unintentionally bias their relative abundance estimates towards hunting areas with more surveys. Now this may not seem like an issue at first glance, since total population sizes likely scale with habitat area overall. However, population density does not necessarily scale with habitat area. Using the marten as an example, we get lower scores using the current ‘totals’ approach (CRE: 1, DBO: 1, GUA: 4, MEN: 1, ROB: 1, SER: 1, SFU: 2) compared to using the alternative ‘averages’ approach (CRE:2, DBO: 4, GUA: 4, MEN: 1, ROB: 1, SER: 1, SFU: 1). Why might this matter? For starters, the results of the author’s analyses will clearly be different (I cannot say how different though, perhaps only a little, perhaps a lot). This goes for both the distance analysis and simple comparisons such as L. 363-364. Second, the answers people provide during interview surveys are likely to be influenced by what they experience during their daily lives. Thus, we would expect there to be some spatial heterogeneity in how people score a species’ abundance. In other words, if someone spends most of their time in an area with many mongooses (i.e., a high local density), then they will probably rate mongooses as being very common even if that species is rare across the landscape overall. On this final note, why do the authors not present any results summarizing (1) interview-derived abundances broken-down by hunting area or (2) distance results broken-down by hunting area? Did the people in each hunting area differ in how they ranked a species’ abundance? Currently, the authors only compare hunters to non-hunters, and this may leave the reader wanting.

The scoring system used to score interview-based abundances is similarly arbitrary and the results will also probably be sensitive to the scoring system used.

Regarding the analyses. The use of Gower’s distance is an interesting choice considering the two abundance indices are (1) on relative scales and thus not directly comparable and (2) and their abundance values are sensitive to the rules used to assign abundance scores (which I noted previously). I think a rank-based correlation analysis (Spearman/Kendall) would overcome (1) and be more robust to (2).

Was there any link between perceived abundance and attitude? In the abstract the authors mention an inverse relationship between relative abundance and human attitude, but I did not see this comparison during my reading of the manuscript. This comparison would be interesting, but I’m guessing there is little to no relationship here considering 93% of respondents said hunting was compatible with conservation and 86% agreed with mesocarnivor control. So I am confused by the statement in the abstract.

Reviewer #2: This is an excellent paper and I am pleased to have been asked to review it. The level of prejudice surmised from the data is consistent with what has been reported elsewhere, namely in North America. I have a few comments that the authors may want to address:

Line 177 - "local people were people inhabiting villages nearby the hunting areas". There could be a conflict of interest in these people's responses due to economic advantages associated with hunting. This might explain (in part) why the respondents considered that hunting was compatible with nature conservation (line 414).

Lines 185-186 - Was consent given before answering the questions? If it was given or withheld afterward, then there is possibility of a bias. For example, if someone reviews the questions and realizes he/she cannot answer them, he/she may withhold his/her consent. So, the lack of knowledge may be higher than what is reported by the researchers.

Statistical methods - what was the level of significance used? 0.05?

Data reporting - Some graphic representation of the main results would be interesting, e.g., a comparison of hunters vs respondents, a comparison of observed vs expected data, etc.

Line 414 - "respondents considered that hunting was compatible with nature conservation" = I think that the proportion of people agreeing with this conclusion would change if the survey for non-hunting respondents had been conducted in an area far from the hunting grounds, e.g. Madrid. However, the lack of ability in recognizing wildlife species may have been the same or lower.

Lines 456-458 - This leads me to believe that the development of management programs should b conducted by wildlife professionals (namely biologists) who are aware of socio-economical realities.

6. PLOS authors have the option to publish the peer review history of their article (what does this mean?). If published, this will include your full peer review and any attached files.

Reviewer #1: No

Reviewer #2: No

---

## [Author Response · Author response to Decision Letter 0]

21 Feb 2023

Please, see the attached Response to Reviewers.docx file.

1. Additional Editor Comments:

Two reviewers have perused the manuscript and differ widely in their recommended decision (Reject vs. Accept). Reviewer 1 in particular has excellent comments and suggestions. I therefore invite you to address all the comments of the reviewers as well as my comments below in a revised manuscript.

We thank the criticisms and the many constructive comments concerning our manuscript provided by the Editor and both reviewers.

Please, find our point-by-point detailed responses below. 

2. Some of the targeted taxa are not mesocarnivore - they are small carnivores (e.g., de Satge et al. 2017 Oecologia 184(4):873-884). They should be classified accordingly throughout the manuscript.

We thanks the reviewer for this comment. De Satgé et al. (2017) certainly considered species such as the common genet (Genetta genetta), and species phylogenetically related to the Egyptian mongoose (Herpestes ichneumon), i.e. the yellow mongoose (Cynictis penicillata), as small carnivores. However, it should be taken into account that research involving our target species in Mediterranean regions usually consider these species as mesocarnivores (e.g., Santos et al. 2016; Curveira-Santos et al. 2017; Ferreras et al. 2017; Descalzo et al. 2020). To be consistent, therefore, we consider maintaining our target species as mesocarnivores instead of small carnivores.

References:

Curveira-Santos, G., T.A. Marques, M. Björklund, and M. Santos-Reis. 2017. Mediterranean mesocarnivores in spatially structured managed landscapes: community organisation in time and space. Agriculture, Ecosystems & Environment 237: 280–289. doi: 10.1016/j.agee.2016.12.037

de Satgé, J., K. Teichman, and B. Cristescu. 2017. Competition and coexistence in a small carnivore guild. Oecologia 184: 873–884 (eng). doi: 10.1007/s00442-017-3916-2

Descalzo, E., J.A. Torres, P. Ferreras, and F. Díaz-Ruiz. 2020. Methodological improvements for detecting and identifying scats of an expanding mesocarnivore in south-western Europe. Mammalian Biology. doi: 10.1007/s42991-020-00062-6

Ferreras, P., F. Díaz-Ruiz, P.C. Alves, and P. Monterroso. 2017. Optimizing camera-trapping protocols for characterizing mesocarnivore communities in south-western Europe. Journal of Zoology 301: 23–31. doi: 10.1111/jzo.12386

Santos, M.J., L.M. Rosalino, M. Santos-Reis, and S.L. Ustin. 2016. Testing remotely-sensed predictors of meso-carnivore habitat use in Mediterranean ecosystems. Landscape Ecology 31: 1763–1780. doi: 10.1007/s10980-016-0360-3

3. European rabbit vs. Iberian hare - can you tell the difference from tracks?

Differences in foot size and distance between feet during movement are clear between rabbit and Iberian hare, being rabbit tracks (1.5-2.0x3.0-4.0 cm) significantly smaller than hare tracks (3.0-3.5x5.0-6.0 cm). Only wet, clay soil surfaces, hinders species identification due to the overprinting of rabbit tracks. In addition, rabbit tracks are usually found in a large number, while hare tracks are usually found composing small groups or alone, given their solitary behaviour. Finally, they are also considered additional factors such as the specific location and the environmental context, which help with track identification. Nevertheless, as we stated in the Materials and methods section of the manuscript, any doubtful assignment of signs to species was discarded to avoid false positives.

Morphometric references taken from:

Sanz Navarro, B. 2012. Huellas y rastros de los mamíferos de la Península Ibérica. Muskari, 210 pp.

4. I disagree that no studies have linked social and ecological data on carnivore distribution in conflict areas. There are many examples of socio-ecological systems and using traditional knowledge in conjunction with species distribution modelling for conflict-prone carnivores. Please add some citations to this effect.

Our main aim was not to evaluate the level of agreement between general traditional knowledge and species distribution models in conflict areas. Instead, we specifically compare the mesocarnivore relative abundance obtained from field studies, carried out in situ, and the collection of individual perception of species distribution through questionnaires made in localities around the surveyed hunting areas. 

During our comprehensive bibliographic search, we did not find examples of investigations addressing a similar question. Nevertheless, we would be pleased to include in the manuscript any possible reference provided by the editor or the reviewers.

5. It would be very useful to add information on size of hunting areas and survey effort in a table.

Many thanks for this comment. We agree that this additional information could be of interest for potential readers of our manuscript. The suggested table including the size of hunting areas and the survey effort is now added in the Table 1 (lines 211-213 of the current version of the manuscript).

6. The calculation of the relative abundance does not seem to be the most appropriate - see for example comments from Reviewer 1. Please also consider changing the standardization of indices (see Reviewer 1's recommendations). Even better, if the data allow an occupancy analysis approach (incorporating detection probability), please replace the relative abundance indices calculations with occupancy analysis. A less favored alternative would be to keep the relative abundance indices, but to acknowledge their limitations.

Regarding to the calculation of the relative abundance, please, see our response to comment 16 below. 

7. Records should not be pooled across field methods (e.g., in Table 2).

We pooled the number of tracks, scats and occasional sightings registered due to practical reasons. Specifically, we combined the records in order not to include in the main text of the manuscript such a specific and large table. 

However, we now provide new supporting information, S8 Table, which comprehensively describes all the records registered for every species, using each field method.

8. Add scale bar (km) to panels in Fig. S2. This figure is important as it shows the sampling units and should be moved to the main manuscript. It shows a gradient of habitat types - from relatively intact habitats to agricultural areas - have you considered incorporating habitat types in the analysis as this will influence carnivore distribution?

We now added the scale bar to panels in Fig S2 and the figure has been moved to the main text. The caption of new Fig 2 is now included in lines 206-210 of the current version of the manuscript and the new figure was also submitted.

Considering the relationship between habitat type and species distribution, from the beginning, as we described in lines 123-130 from the original manuscript (lines 114-119 from the current version), we assumed that the different hunting areas represented different habitat types, which certainly would influence species distribution. 

However, our main aim was not to analyse species-habitat relationships, as there is a vast body of literature addressing this objective (e.g. Goldyn et al. 2003, Byrne et al. 2012), also in Mediterranean regions (e.g. Palomares and Delibes 1990, Santos and Santos-Reis 2010, Camps and Alldredge 2013). Instead, we wanted to analyse the degree of similarity between the perceived and the measured relative abundance to unveil whether attitude towards carnivores is underpinned or it might be biased by alternative factors. 

Hence, we accounted for this variation in species distribution across habitat types specifically using GLMM in our analysis, controlling for the environmental heterogeneity associated with the hunting areas, considering Hunting area as a random variable. We stated this in lines 225-227 from the original manuscript, and it is also highlighted in lines 229-231 from the current version.

References:

Byrne, A.W., D. Paddy Sleeman, J. O'Keeffe, and J. Davenport. 2012. The ecology of the European badger (Meles meles) in Ireland: a review. Biology & Environment: Proceedings of the Royal Irish Academy 112: 105–132. doi: 10.3318/BIOE.2012.02

Camps, D., and J.R. Alldredge. 2013. Multi-scale habitat use and selection of common genet Genetta genetta (Viverridae, Carnivora) in a Mediterranean environment. Mammalia 77: 285–295. doi: 10.1515/mammalia-2012-0023

Goldyn, B., M. Hromada, A. Surmacki, and P. Tryjanowski. 2003. Habitat use and diet of the red fox Vulpes vulpes in an agricultural landscape in Poland. Z. Jagdwiss. 49: 191–200.

Palomares, F., and M. Delibes. 1990. Habitat preference of large grey mongoose Herpestes ichneumon in Spain. Acta Theriologica 35: 1–6.

Pandolfi, M., P. Forconi, and L. Montecchiari. 1997. Spatial behaviour of the red fox (Vulpes vulpes) in a rural area of central Italy. Italian Journal of Zoology 64: 351–358. doi: 10.1080/11250009709356222

Santos, M.J., and M. Santos-Reis. 2010. Stone marten (Martes foina) habitat in a Mediterranean ecosystem: effects of scale, sex, and interspecific interactions. European Journal of Wildlife Research 56: 275–286. doi: 10.1007/s10344-009-0317-9

9. Fig. S4 (panel A) is also important to include in the main manuscript.

Done. The caption of new Fig 3 is now added in lines 355-356 of the current version of the manuscript and the new figure was also submitted.

10. Please credit the background satelite image in Fig. 1.

Done. Please see our response to comment 4.

11. Reviewer #1: I have reviewed the manuscript entitled: “Social perception of mesocarnivores within hunting areas differs from actual species abundance”. The manuscript covers a very important topic in wildlife management and conservation. I have several concerns regarding the underlying assumptions and methodology though. These concerns are raised below.

For the manuscript in question to make the claims it does, we need to know how reliable the so-called “real abundance” estimates are. This is because the study does not aim, for example, to compare two methods of detecting a species’ presence, but instead aims to examine how closely people’s perceptions of a species’ abundance match reality. Understandably, robust estimates of population abundance or density are not feasible for these species. Therefore, the use of a proxy is required. For the study in question, the authors use the number of detections (track, scat, or visual/opportunistic) as the basis for their relative abundance index. However, relative abundance indices that do not account for imperfect detection are known to be flawed measures that may produce biased or even spurious results. Since the underlying thesis of this study depends on the reliability of the field-sampling based abundance estimates, one must appreciate this limitation severely hinders the manuscript in question. On this criticism alone I would say the manuscript is dead-on-arrival. If I may ask, are the field data wholly unsuitable for occupancy analysis? Using occupancy probabilities derived from a model that accounts for imperfect detection to rank-order species would be better.

According to MacKenzie (2006), in its most simple and fundamental form, occupancy modelling considers a basic sampling scheme in which s sites are each surveyed K times for the target species. During each survey, appropriate methods are employed to detect the species at the sites. Such methods include the direct or indirect confirmation of at least one individual of the species, assuming that the target species are never falsely detected at a site when absent (i.e., by misidentification of the species), which is likely to be a reasonable assumption in many situations (MacKenzie 2006).

Nevertheless, due to the lack of sufficient of financial resources and the limitations derived from the movement restrictions due to the COVID-19 crisis, we only carried on one single visit into a limited number of sites in which we obtain permission to carry out the surveys. Currently, there are some estimations of occupancy that employs single-visit data, but unbiased and precise estimates need demanding requirements such as a vast number of sites (Peach et al, 2017, Lauret et al. 2021). 

However, with regard to the reliability of field-sampling methods, in the Materials and methods section we provide two specific references in which two main ideas are exposed: the reliability of field methods employed (Suárez-Tangil and Rodríguez 2021a) and their uniform performance despite variations in environmental conditions (Suárez-Tangil and Rodríguez 2021b). 

On the one hand, Suárez-Tangil and Rodriguez (2021a) provided a comprehensive assessment of multiple detection methods, including track and scat surveys. This assessment included a comparison of the probability of detection for target mammal species (which comprised the five mesocarnivores and two of the three small game species studied in the present manuscript). Specifically, in Figure 3, authors provide the mean (±SE) probability of detection for target mammal species estimated with competitive occupancy models. 

According to their results, mean probability of detection for track and scat surveys were notably higher than for the rest of the sampling methods employed for red fox, common genet, Egyptian mongoose, Eurasian badger and stone marten. Additionally, they showed that the estimated probability of detection of track and scat surveys were similar for the rabbit and the Iberian hare.

On the other hand, Suárez-Tangil and Rodríguez (2021b) examined the influence of spatio-temporal factors on the performance of survey methods, concluding that i) after controlling for variations in mammal relative abundance or activity associated with landscape and season, the effect of spatio-temporal factors on the performance of detection methods was small; and ii) track surveys proved to be the most efficient and fastest method for detecting mammal species, being barely influenced by spatio-temporal factors.

Both investigations were carried out in Mediterranean regions with similar land cover (agricultural landscapes and restricted natural vegetation areas), and comprised most of the target species studied in our manuscript. Therefore, we considered that their conclusions are applicable to our manuscript, and that the reliability of the field-sampling methods might not hinder the manuscript at all.

References: 

Lauret, V., H. Labach, M. Authier, and O. Gimenez. 2021. Using single visits into integrated occupancy models to make the most of existing monitoring programs. Ecology 102: e03535 (eng). doi: 10.1002/ecy.3535

MacKenzie, D.I. 2006. Occupancy estimation and modeling: Inferring patterns and dynamics of species. Amsterdam, Boston: Elsevier, xviii, 324.

Peach, M.A., J.B. Cohen, and J.L. Frair. 2017. Single-visit dynamic occupancy models: an approach to account for imperfect detection with Atlas data. Journal of Applied Ecology 54: 2033–2042. doi: 10.1111/1365-2664.12925

Suárez-Tangil BD, Rodríguez A. Integral assessment of active and passive survey methods for large-scale monitoring of mammal occurrence in Mediterranean landscapes. Ecological Indicators. 2021a;125, 107553. https://doi.org/10.1016/j.ecolind.2021.107553

Suárez‐Tangil BD, Rodríguez A. Uniform performance of mammal detection methods under contrasting environmental conditions in Mediterranean landscapes. Ecosphere. 2021b;12(3), e03349. https://doi.org/10.1002/ecs2.3349

12. Moving beyond the quality of the data, there is the issue of how the authors produced the relative abundance scores. The approach used seems arbitrary and I can imagine the study’s results will probably be sensitive to the scoring system used. Currently, the authors sum up the number of detections within a given hunting area and then divide each area’s total by the total number of detections within hunting area with the most detections. I imagine the authors could just as easily divide each hunting area’s total by the total number of detections from all hunting areas combined and that would presumably yield different results (though perhaps not that different since most detections come from one or two hunting areas). Likewise, the authors could easily average the number of detections across surveys within each hunting area and use hunting area averages instead of hunting area totals. I’m sure we can think up countless ways to score a species’ relative abundance, but I bring up the use of averages specifically for a reason. By totaling the number of detections across multiple surveys, the authors unintentionally bias their relative abundance estimates towards hunting areas with more surveys. Now this may not seem like an issue at first glance, since total population sizes likely scale with habitat area overall. However, population density does not necessarily scale with habitat area. Using the marten as an example, we get lower scores using the current ‘totals’ approach (CRE: 1, DBO: 1, GUA: 4, MEN: 1, ROB: 1, SER: 1, SFU: 2) compared to using the alternative ‘averages’ approach (CRE:2, DBO: 4, GUA: 4, MEN: 1, ROB: 1, SER: 1, SFU: 1). Why might this matter? For starters, the results of the author’s analyses will clearly be different (I cannot say how different though, perhaps only a little, perhaps a lot). This goes for both the distance analysis and simple comparisons such as L. 363-364. 

In order to analyse the similarity between field sampling and people perception we needed to make figures comparable. Whilst not denying that there might be countless (or at least a few more) ways to score a species’ relative abundance, our procedure was not arbitrary, but limited by two principal reasons. 

On the one hand, we needed to simplify the potential answer on perceived abundance and attributed damage, as it would have been hard for the respondents to pin down a range number for the species relative abundance or to provide a wider range of possibilities to choose, and make the score comparable. 

On the other hand, we did not employ the average number of detections across surveys within each Hunting area, as suggested by the reviewer, because we did not want to consider each sampling unit as independent for the score estimation. In our opinion, employing the average would have provided an undesired weight to each sampling unit in the analyses, and what we wanted was to consider the Hunting area as a whole, given that people perception is not fundamentally based on particular species abundance in specific sampling units, but on the whole area. Therefore, we established an index of relative abundance by calculating the proportion of records found in each hunting area with respect to the maximum observed value (i. e. the maximum number of records found in a hunting area). 

13. Second, the answers people provide during interview surveys are likely to be influenced by what they experience during their daily lives. Thus, we would expect there to be some spatial heterogeneity in how people score a species’ abundance. In other words, if someone spends most of their time in an area with many mongooses (i.e., a high local density), then they will probably rate mongooses as being very common even if that species is rare across the landscape overall. On this final note, why do the authors not present any results summarizing (1) interview-derived abundances broken-down by hunting area or (2) distance results broken-down by hunting area? Did the people in each hunting area differ in how they ranked a species’ abundance? Currently, the authors only compare hunters to non-hunters, and this may leave the reader wanting.

We really appreciate this comment and certainly, this is an interesting point.

To address this issue, in early stages of the development of the statistical analyses, we studied the potential relationship between both the perceived abundance and the attributed damage and the Hunting area. 

Regarding the perceived abundance, we carried out Pairwise comparisons using Tukey and Kramer (Nemenyi) test with Tukey-Dist approximation for independent samples. We found that there were no consistent differences across Hunting areas (please see R1 Table below, where we describe the number of pairwise comparisons between hunting areas in which we found significant differences in the perceived abundance of carnivore species, p < 0.05). Instead, we found a general pattern with non-significant differences in the perceived abundance, regardless the Hunting area. Specifically, from 27 pairwise comparisons (7 hunting areas), the case in which we found the highest number of significant differences associated with the Hunting area was 5, corresponding to the perceived abundance of the common genet. Nevertheless, we found a great variation in people’s responses and, therefore, acknowledge that a higher sample size would provide more robust results. 

Non-consistent differences were especially remarkable for the red fox and the Egyptian mongoose (please see R1 Fig A and R1 Fig B below), which were perceived as very abundant across all Hunting areas, and for the stone marten, which was perceived as very rare across all Hunting areas (R1 Fig C). In the case of the Eurasian badger, we found that in most cases, it was considered as relatively abundant across Hunting areas (R1 Fig D) and, finally, in the case of the common genet, it was considered as very rare in most of Hunting areas (R1 Fig E). 

Species Pairwise comparison

Red fox 0

Egyptian mongoose 1

Stone marten 4

Eurasian badger 4

Common genet 5

As we concluded that the perceived abundance was more related to individual perceptions and biases than to the spatial context in which people lives, we only introduced the Hunting area as a random variable in the GLMM to control for the environmental heterogeneity, instead of consider it as a covariate affecting people’s perception. 

14. The scoring system used to score interview-based abundances is similarly arbitrary and the results will also probably be sensitive to the scoring system used.

Please see our response to comment 17.

15. Regarding the analyses. The use of Gower’s distance is an interesting choice considering the two abundance indices are (1) on relative scales and thus not directly comparable and (2) and their abundance values are sensitive to the rules used to assign abundance scores (which I noted previously). I think a rank-based correlation analysis (Spearman/Kendall) would overcome (1) and be more robust to (2).

We appreciate reviewer’s suggestion. 

However, we decided to applied Gower’s distance because in case of discrete, numerical values such as the scores calculated for both field sampling data and species perceived abundance, Gower’s distance is adequately applied, given that distance is calculated as the absolute value of the difference between two records (similar to Manhattan distance), which is precisely our aim.

16. Was there any link between perceived abundance and attitude? In the abstract the authors mention an inverse relationship between relative abundance and human attitude, but I did not see this comparison during my reading of the manuscript. This comparison would be interesting, but I’m guessing there is little to no relationship here considering 93% of respondents said hunting was compatible with conservation and 86% agreed with mesocarnivor control. So I am confused by the statement in the abstract.

With that sentence in the abstract, we aimed to succinctly highlight that, in general, for both hunters and other local people, the value of mesocarnivores perceived abundance was virtually opposite to that provided by the field sampling. 

We rephrase the sentence in the abstract to clarify (lines 32-33 from the current version of the manuscript). 

17.Reviewer #2: This is an excellent paper and I am pleased to have been asked to review it. The level of prejudice surmised from the data is consistent with what has been reported elsewhere, namely in North America. I have a few comments that the authors may want to address:

Thank you very much. We are glad the reviewer was pleased to have been asked to review our manuscript.

We provide below responses to all comments. 

18. Line 177 - "local people were people inhabiting villages nearby the hunting areas". There could be a conflict of interest in these people's responses due to economic advantages associated with hunting. This might explain (in part) why the respondents considered that hunting was compatible with nature conservation (line 414).

That is an interesting point. 

What we wanted when we were specifically looking for people nearby the hunting area was, firstly, to control the potential lack of knowledge in people living far from where species occur and, secondly, to extract information from people whose livelihood was directly or indirectly linked with hunting, e.g. due to financial or recreational reasons.

What we perceived in most of the interviews was that there was an inherent commitment in favour of nature conservation, which partially, or even totally, clashed with classical ideas associated with hunting, for example the family tradition, the wealth of local small businesses, or the control of mesocarnivores to mitigate the potential harm caused to poultry and small game species.

19. Lines 185-186 - Was consent given before answering the questions? If it was given or withheld afterward, then there is possibility of a bias. For example, if someone reviews the questions and realizes he/she cannot answer them, he/she may withhold his/her consent. So, the lack of knowledge may be higher than what is reported by the researchers. 

Before we conduct each survey and, therefore, make any question, we specifically asked the respondents if they wanted to answer some questions for a study related with people’s perception of carnivores and small game species known to be present in the area, and their relationship with hunting, particularly in the hunting areas located nearby the village. Additionally, despite the respondents provided us their consent before answering the questions, we did not force any respondent to answer to any question. If the respondent was not comfortable with the question, for example because it could prove their lack of knowledge or they thought that some particular answers might show a political view they did not want to share for some reason, we gave them the opportunity of not responding.

20. Statistical methods - what was the level of significance used? 0.05?

That is correct, we used a level of significance of 0.05 in the analyses.

21. Data reporting - Some graphic representation of the main results would be interesting, e.g., a comparison of hunters vs respondents, a comparison of observed vs expected data, etc.

Following the suggestion provided by both the editor and the reviewer, we added a new table, Table 4 in the current version of the manuscript (lines 393-400), which shows the dissimilarity between the relative abundance perceived by respondents and the relative abundance recorded during field sampling.

Additionally, we have added specific references to supporting information across the manuscript (including some graphic representations), which may also help to report our results.

22. Line 414 - "respondents considered that hunting was compatible with nature conservation" = I think that the proportion of people agreeing with this conclusion would change if the survey for non-hunting respondents had been conducted in an area far from the hunting grounds, e.g. Madrid. However, the lack of ability in recognizing wildlife species may have been the same or lower.

Please see our response to comment 23.

23. Lines 456-458 - This leads me to believe that the development of management programs should b conducted by wildlife professionals (namely biologists) who are aware of socio-economical realities.

In our opinion, this is one of the principal take-home messages of our manuscript, as we stated a few lines below (lines 486-500 of the current version of the manuscript):

“Our results may provide valuable information for the design of future management strategies, as they describe three interesting aspects that may be overlooked if social perception and objective evidences are considered independently when dealing with human-carnivore conflicts: disparities between perceived and actual species abundance, biases in social perception derived from knowledge gaps in species identification, and positive willingness to harmonize hunting and nature conservation.

[…]

Nevertheless, our study clearly indicates that social perception should be considered in conjunction with a comprehensive knowledge of species distribution in order to avoid unjustified bias and adopt informed and appropriate decisions in the management of human-carnivore conflicts.”

---

## [Editor Report · Decision Letter 1]

3 Mar 2023

PONE-D-22-29695R1Social perception of mesocarnivores within hunting areas differs from actual species abundancePLOS ONE

Dear Dr. Suárez de Tangil,

Thank you for submitting your manuscript to PLOS ONE. After careful consideration, we feel that it has merit but does not fully meet PLOS ONE’s publication criteria as it currently stands. Therefore, we invite you to submit a revised version of the manuscript that addresses the points raised during the review process.

We look forward to receiving your revised manuscript.

Kind regards,

Bogdan Cristescu

Academic Editor

PLOS ONE

Journal Requirements:

Academic Editor Comments (if provided):

Most comments have been addressed. I recommend the following revisions:

Abstract:

I suggest replacing as follows:

"social perception and field data"

social perception and ecological field data

"might help identifying"

might help identify

"potential biases in human-carnivore conflicts"

potential biases in identifying and managing human-carnivore conflicts

"The main objective of this work is to analyse"

We analyzed

"perceived and the measured relative abundance"

perceived and field-measured relative abundance of carnivores

"mesocarnivores perceived abundance, for both stakeholders was quite different to actual species abundance"

mesocarnivore perceived abundances were generally different to actual species abundances

"respondents’ ability to identify the species"

respondents’ ability to identify the carnivore species

"the need of increase people knowledge of species distribution and characteristics before adopting decisions when managing human-carnivore conflicts, especially for stakeholders that are directly involved in"

the need to increase people's knowledge on species distribution and ecological characteristics before adopting decisions when managing human-carnivore conflicts

Keywords:

Add some keywords for methods used (e.g., track survey etc.)

Table 2 caption:

"capacity of mesocarnivore identification"

I suggest replacing as follows:

ability to identify mesocarnivores

Table 3:

Are the values in the table standardized by km2? They should be

In your response to Reviewer 1, you state:

"On the other hand, Suárez-Tangil and Rodríguez (2021b) examined the influence of spatiotemporal factors on the performance of survey methods, concluding that i) after controlling for variations in mammal relative abundance or activity associated with landscape and season, the effect of spatio-temporal factors on the performance of detection methods was small; and ii) track surveys proved to be the most efficient and fastest method for detecting mammal species, being barely influenced by spatio-temporal factors. Both investigations were carried out in Mediterranean regions with similar land cover (agricultural landscapes and restricted natural vegetation areas), and comprised most of the target species studied in our manuscript. Therefore, we considered that their conclusions are applicable to our manuscript, and that the reliability of the field-sampling methods might not hinder the manuscript at all".

Please condense the information above and include it in the manuscript to highlight your assumption described above.
---

## [Author Response · Author response to Decision Letter 1]

3 Mar 2023

1. Academic Editor Comments (if provided):

Most comments have been addressed. I recommend the following revisions:

Abstract:

I suggest replacing as follows:

"social perception and field data"

social perception and ecological field data.

Text modified as suggested by the Editor (line 27 of the current version of the manuscript).

2. "might help identifying"

might help identify

Done (line 27 of the current version of the manuscript).

3. "potential biases in human-carnivore conflicts"

potential biases in identifying and managing human-carnivore conflicts

The sentence was rephrased as suggested by the editor (line 28 of the current version of the manuscript).

4. "The main objective of this work is to analyse"

We analyzed

The sentence was rephrased as suggested by the editor (line 28 of the current version of the manuscript).

5. "perceived and the measured relative abundance"

perceived and field-measured relative abundance of carnivores

The sentence was partially modified, given that field-measured abundance also includes small game species (line 29 of the current version of the manuscript).

6. "mesocarnivores perceived abundance, for both stakeholders was quite different to actual species abundance"

mesocarnivore perceived abundances were generally different to actual species abundances

The sentence was rephrased as suggested by the editor (lines 32-33 of the current version of the manuscript).

7. "respondents’ ability to identify the species"

respondents’ ability to identify the carnivore species

Done (lines 34-35 of the current version of the manuscript).

8. "the need of increase people knowledge of species distribution and characteristics before adopting decisions when managing human-carnivore conflicts, especially for stakeholders that are directly involved in"

the need to increase people's knowledge on species distribution and ecological characteristics before adopting decisions when managing human-carnivore conflicts

The sentence was modified (lines 35-37 of the current version of the manuscript).

9. Keywords:

Add some keywords for methods used (e.g., track survey etc.)

We removed the expressions ‘attitude towards carnivores’ and ‘field sampling’ from Keywords and replaced them with ‘track surveys’ and ‘scat surveys’ (line 40 of the current version of the manuscript).

10. Table 2 caption:

"capacity of mesocarnivore identification"

I suggest replacing as follows:

ability to identify mesocarnivores

The sentence was rephrased (lines 315-316 of the current version of the manuscript).

11. Table 3:

Are the values in the table standardized by km2? They should be

Values in Table 3 represent the relative abundance of mesocarnivores and small game species found in each 4-km2 sampling unit surveyed. However, following editor’s suggestion, we now provide, marked in bold, the total relative abundance standardized by hunting area surface in km2 (lines 375-377 of the current version of the manuscript).

12. In your response to Reviewer 1, you state:

"On the other hand, Suárez-Tangil and Rodríguez (2021b) examined the influence of spatiotemporal factors on the performance of survey methods, concluding that i) after controlling for variations in mammal relative abundance or activity associated with landscape and season, the effect of spatio-temporal factors on the performance of detection methods was small; and ii) track surveys proved to be the most efficient and fastest method for detecting mammal species, being barely influenced by spatio-temporal factors. Both investigations were carried out in Mediterranean regions with similar land cover (agricultural landscapes and restricted natural vegetation areas), and comprised most of the target species studied in our manuscript. Therefore, we considered that their conclusions are applicable to our manuscript, and that the reliability of the field-sampling methods might not hinder the manuscript at all".

Please condense the information above and include it in the manuscript to highlight your assumption described above.

We added the following sentence, written in lines 192-194 of the current version of the manuscript:

‘The regular performance and high efficiency of track and scat surveys have been proven in other Mediterranean regions with similar land covers’.

---

## [Editor Report · Decision Letter 2]

20 Mar 2023

Social perception of mesocarnivores within hunting areas differs from actual species abundance

PONE-D-22-29695R2

Dear Dr. Suárez de Tangil,

We’re pleased to inform you that your manuscript has been judged scientifically suitable for publication and will be formally accepted for publication once it meets all outstanding technical requirements.

Kind regards,

Bogdan Cristescu

Academic Editor

PLOS ONE

Additional Editor Comments (optional):

Congratulations on your paper. I have one comment that you can address at the proofs stage. Please ensure that the data in Table 3 is standardized per 4 km2 sampling unit. Modify the caption of Table 3 to make it clear that the data is presented per standardized sampling unit (4 km2). The caption of that table was a bit confusing, hence my earlier recommendation to ensure standardization. Reporting the data in the table per 4 km2 sampling unit, and specifying clearly in the table caption that reporting was done per 4 km2, will ensure clarity.

---

## [Editor Report · Acceptance letter]

19 Apr 2023

PONE-D-22-29695R2 

Social perception of mesocarnivores within hunting areas differs from actual species abundance 

Dear Dr. Suárez de Tangil:

I'm pleased to inform you that your manuscript has been deemed suitable for publication in PLOS ONE. Congratulations! Your manuscript is now with our production department. 

Kind regards, 

on behalf of

Dr. Bogdan Cristescu 

Academic Editor

PLOS ONE